# Clinical factors associated with rapid treatment of sepsis

Xing Song[1]*, Mei Liu[2], Lemuel R. Waitman[1], Anurag Patel[3], Steven Q. Simpson[4]*

1 Health Management and Informatics, School of Medicine, University of Missouri, Columbia, MO, United States of America, 2 Division of Medical Informatics, Department of Internal Medicine, University of Kansas Medical Center, Kansas City, KS, United States of America, 3 Anurag4Health, Kansas City, KS, United States of America, 4 Pulmonary and Critical Care Division, Department of Internal Medicine, University of Kansas Medical Center, Kansas City, KS, United States of America

* ssimpson3@kumc.edu (SQS); xsm7f@health.missouri.edu (XS)

## Abstract

### Purpose

To understand what clinical presenting features of sepsis patients are historically associated with rapid treatment involving antibiotics and fluids, as appropriate.

### Design

This was a retrospective, observational cohort study using a machine-learning model with an embedded feature selection mechanism (gradient boosting machine).

### Methods

For adult patients (age ≥ 18 years) who were admitted through Emergency Department (ED) meeting clinical criteria of severe sepsis from 11/2007 to 05/2018 at an urban tertiary academic medical center, we developed gradient boosting models (GBMs) using a total of 760 original and derived variables, including demographic variables, laboratory values, vital signs, infection diagnosis present on admission, and historical comorbidities. We identified the most impactful factors having strong association with rapid treatment, and further applied the Shapley Additive exPlanation (SHAP) values to examine the marginal effects for each factor.

### Results

For the subgroups with or without fluid bolus treatment component, the models achieved high accuracy of area-under-receiver-operating-curve of 0.91 [95% CI, 0.86–0.95] and 0.84 [95% CI, 0.81–0.86], and sensitivity of 0.81[95% CI, 0.72–0.87] and 0.91 [95% CI, 0.81–0.97], respectively. We identified the 20 most impactful factors associated with rapid treatment for each subgroup. In the non-hypotensive subgroup, initial physiological values were the most impactful to the model, while in the fluid bolus subgroup, value minima and maxima tended to be the most impactful.

**Data Availability Statement:** Data cannot be shared publicly because it includes high-dimensional patient-level data. The high-dimensionality nature of this dataset has a potential re-identification risk if linked with other data

source. Data are available from the University of Kansas Medical Center Institutional Data Access / Ethics Committee (contact via HERON team at University of Kansas Medical Center, misupport@kumc.edu, for researchers who meet the criteria for access to confidential data).

**Funding:** SQS and AP received Blue KC Outcome Research Grants (No.0925-0001) and the authors played role in study design, decision to publish and preparation of the manuscript. LRW received CTSA grant UL1TR002366 from NCRR/NIH and the author played role in data collection and preparation of the manuscript. XS played role in study design, developed the training and testing setup, extracted the study cohort, cleaned up the data and performed all experiments ML played role in study design and manuscript revision. The funders provided support in the form of salaries for authors XS, ML, LRW, AP, SQS, but did not have any additional role in the study design, data collection and analysis, decision to publish, or preparation of the manuscript. The specific roles of these authors are articulated in the 'author contributions' section as well as above.

**Competing interests:** AP's affiliation to Anurag4Health does not alter our adherence to PLOS ONE policies on sharing data and materials.

**Abbreviations:** SSC, Surviving Sepsis Campaign; EMR, Electronic Medical Record; SIRS, Systemic Inflammatory Response Syndrome; ED, Emergency Department; HERON, Healthcare Enterprise Resource for Ontological Narration; EMS, Emergency Medical Service; GBM, Gradient Boosting Machine; SHAP, Shapley Additive exPlanation values; AUROC, Area Under the Receiver Operator Characteristic curve; AUPRC, Area Under the Precision Recall Characteristic curve; PPV, positive predictive value; HL, Hosmer-Lemeshow test; SBP, Systolic Blood Pressure; DBP, Diastolic Blood Pressure; MAP, Mean Arterial Pressure; GCS, Glasgow Coma Scale; LOC, Level of Consciousness; BMP, Basic Metabolic Panel; BUN, Blood Urea Nitrogen; CBC, Complete Blood Count; LFT, Liver Function Test; SCr, Serum Creatinine; WBC, White Blood Cell; LODS, Logistic Organ Dysfunction Score; IQR, Inter-Quartile Range.

## Conclusion

These machine learning methods identified factors associated with rapid treatment of severe sepsis patients from a large volume of high-dimensional clinical data. The results provide insight into differences in the rapid provision of treatment among patients with sepsis.

## Introduction

Sepsis is an important public health problem in the United States and is the leading cause of death among hospitalized patients. Current estimates suggest that sepsis afflicts over 1.7 million Americans each year and is responsible for over 270,000 deaths [1]. In 2012 the Surviving Sepsis Campaign (SSC) first published 3-hour and 6-hour care bundles for reducing mortality due to sepsis [2]. Based on these recommendations, hospitals implement the guidelines differently, adapted to local standards of care, and clinicians behave differently, based on patients' manifestations of illness. Education of healthcare workers and attention to quality improvement have aided in reducing mortality from sepsis [1, 2]. To that end, numerous efforts have used alerting mechanisms within the electronic medical record (EMR) to attempt early warning of the signs of sepsis, whether based on systemic inflammatory response syndrome (SIRS) or organ dysfunction, with varying degrees of success [3–5]. One weakness of EMR-based alerting has been its inability to detect when infection is suspected or present, while the strengths of the approach lie in identifying objective data, such as respiratory rate, blood pressure, or specific criteria of organ dysfunction in sepsis [6].

EMRs have revolutionized the curation and presentation of clinical data; in the current state, medicine is far better served than it has ever been, in terms of having data readily available for clinical use. However, the mode of presentation of data to end users (physicians, nurses, etc.) remains steadfastly in a 20th century paradigm, in that EMRs predominantly exist as information storehouses, and their potential to guide efficient diagnosis and treatment decisions for various conditions is unrealized, as is their potential to facilitate quality improvement efforts. EMRs currently provide users with static patient data; the values are displayed in the manner of the specific EMR and are displayed in essentially the same fashion, regardless of the identity of the user. The physician user mostly sees only the data that they have specifically sought, and frequently this data is sought principally to confirm prior beliefs about the patient. We envision an EMR that is adaptive to the patient's data, the location or facility, and the user, much as the consumer-oriented products Amazon and Google are. In other words, the envisioned EMR presents the data to an individual user in a fashion intended to promote specific behaviors and based on adaptive algorithms determined by users' collective previous desired behavior. For Amazon or Google, those desired behaviors involve product purchases or the viewing of specific web pages. For the EMR, such behaviors could include rapid treatment of sepsis.

Development of such an EMR requires several steps: a) understanding which conditions can benefit from it, b) within those conditions, determining the features that are associated with efficient diagnosis and care delivery that alter patient outcomes, c) understanding which of those features are promoters of efficient diagnosis and treatment, rather than simply covariates, d) testing putative promoters by altering modes of EMR display to individual users and monitoring subsequent diagnostic and therapeutic activity, along with patient outcomes. We

chose to study sepsis, as it is a high morbidity and mortality condition that is also the most expensive condition treated in American hospitals [7].

Numerous efforts have been made to improve prognostic accuracy and efficiency for sepsis and its complications via machine learning techniques. For example, Yang et al. [8], Komorowski et al. [9] and Reyna et al. [10] developed artificial intelligence models to predict sepsis in intensive care; while Mao et al. [11] and Lauritsen et al. [12] extended the prediction application to ED and general ward. Itzhak et al. [13] and Cherifa et al. [14] developed models to predict acute hypertensive or hypotensive episodes among ICU admissions. However, few, if any studies have been designed to understand specific clinical features that patients exhibit at the time that physicians initiate rapid treatment of sepsis. Without assuming causality, one could evaluate from a situational awareness perspective which clinical features are most closely associated with rapid, thorough treatment. We believed that data from such a study could provide novel information that could be used to prompt rapid sepsis treatment for appropriate patients, regardless of the extant diagnostic criteria. We performed a retrospective, machine learning analysis of patients presenting to our ED over a ten-year period to identify all patients meeting clinical criteria for severe sepsis, and to determine which of their clinical characteristics were associated with rapid initiation of antibiotics, fluids, and other sepsis treatments.

## Methods

We collected a retrospective, observational cohort of adult patients (age $\geq$ 18 years) who were admitted through the University of Kansas Hospital ED from 11/2007 through 05/2018. The de-identified data were obtained from the Healthcare Enterprise Resource for Ontological Narration (HERON), an i2b2-based clinical integrated data repository [15, 16]. The operation of HERON as an honest broker research repository was approved by the University of Kansas Medical Center Institutional Review Board (Human Subject Committee) as an expedited protocol and is renewed and reviewed annually (HSC #12337).

Because of the date range and the nature of these studies, we used the definitions of sepsis, severe sepsis, and septic shock according to the American College of Chest Physicians/Society of Critical Care Medicine (Sepsis-1) definitions, including the SIRS and the laboratory thresholds for organ dysfunction [17]. Patients were included by satisfying all of the following criteria:

- presence of a suspected infection is based on clinicians' actions to diagnose and treat infection, defined as a body fluid culture ordered and anti-infective administered within four hours of one another;

- presence of two or more SIRS criteria;

- at least one site of acute organ dysfunction, which was defined by the first instance of an abnormal laboratory or examination value and based on organ dysfunction criteria from the first and second international consensus definitions [2, 6].

To further exclude patients who were admitted through the ED but developed sepsis later in their hospitalization, i.e., to include only sepsis present on admission, we inferred an hour boundary based on the timing distribution of patients with infection present on admission, which was 13 hours since triage.

The outcome of interest was timely completion of the SSC 3-hour bundle components [17]. We chose the SSC bundles not as an endorsement, but as quantifiable, time-stamped, and recorded actions that are representative of rapid treatment and that are widely known to critical care and emergency practitioners. The specific treatment bundles were not proposed until 2012, but the components of the 3-hour bundle represent standard elements of excellent sepsis care

and have always been present in well-treated patients. However, during the entire study time period Sepsis-2 definitions were the hospital standard and SSC bundles were promoted via continuous quality improvement. No specific sepsis treatments are initiated in our region by Emergency Medical Service (EMS) personnel. In the ED, only physicians may order antibiotics or fluids, though blood cultures may be initiated by nursing personnel. We defined the responses separately for two subgroups, based on the SSC bundle recommendations: Group 1- if a fluid bolus was never triggered by hypotension (systolic blood pressure < 90 mm/Hg, mean arterial pressure < 70 mm Hg, or documented drop in systolic blood pressure ≥ 40 mm Hg) or lactate ≥ 4 mmol/L was not present, we defined rapid treatment as completion of the remaining bundle components within 3 hours of triage; Group 2—if a fluid bolus was triggered, we defined rapid treatment as Group 1 actions, plus completion of a 2-liter bolus within 2 hours of bolus initiation. Thus, separate feature selection models were developed for each subgroup.

We adopted a gradient boosting machine (GBM), an embedded feature selection technique which performed feature selection while constructing and optimizing a prediction model, on the two subgroups separately [18]. GBM is an ensemble learning technique that generates a sequence of decision trees, each of which is designed to further improve prediction accuracy from the previous trees [19, 20]. We randomly partitioned the data into a training set (70% of patients) for model development and a testing set (30% of patients) used for measuring prediction accuracy. To control overfitting, we carefully tuned the model hyper-parameters (i.e. depth of each tree, number of trees, learning rate, and minimum-child-weight) within the training set using 10-fold cross validation. At each iteration during the training stage, we performed "down-sampling", a common technique of sampling positive and negative cases in equal proportion at each node of each tree to avoid overweighting by negative cases [21]. Missing values were handled in the following fashion: for categorical data, a value of 0 was set for missing whereas for numerical data, a *missing value split* was always accounted for, and the *best* imputation value can be adaptively learned based on improvement in training AUROC, at each tree node within the ensemble. For example, if a variable $X$ takes values (0, 1, 2, 3, NA, and NA), where "NA" stands for missing, the following 2 decisions will be made automatically at each split for each tree: (a) should we split based on missing or not; (b) if we split based on values, for example, > 1 or ≤ 0, should we merge the missing cases with the bin of > 1 or ≤ 0. We used the R package "xgboost" and SHAP value derivation used the "xgboostExplainer" package for model development [22, 23]. Additionally, because the data encompass a crucial decade in the development of sepsis diagnosis and treatment, we evaluated whether treatment year was a significant feature of the data, by stratifying the validation set by year.

Seven hundred sixty distinct predictors were fed into the model, including demographics, vital signs, routine laboratory values, a variety of statistics that summarize vital sign and laboratory trends when multiple observations were made, clinical manifestations of Systemic Inflammatory Response Syndrome (SIRS) and acute organ dysfunction, as well as infection diagnosis present on admission and comorbidity diagnoses before admission (Table 1). To include factors that were more likely to induce rapid treatment, rather than being an outcome of it, we sampled values that occurred strictly before IV fluid initiation for Group 2 patients. For Group 1 patients, we sampled values until all sepsis-defining values were present or until completion of the bundle when that occurred before all sepsis-defining values were present, which we called the prediction point. Factors were selected based on their collective discriminant power, measured by area under the receiver operator characteristic curve (AUROC), and the optimal sensitivity and specificity determined by the point closest to the top-left corner of ROC curve by Euclidean distance. We also evaluated area under the precision recall curve (AUPRC) and the positive predictive value (PPV) when optimal sensitivity was achieved, as well as calibration score measured by Brier score and the Hosmer-Lemeshow test (HL).

**Table 1. Complete list of variables included in the full model.**

| | | |
|---|---|---|
| **Demographics** <br><br> **(4)** | Age, Sex, Race, Ethnicity | Continuous <br> • age <br> Binary <br> • sex, race, ethnicity. |
| **Vital signs** <br><br> **(20)** | Temperature, Heart Rate, Respiratory Rate, Systolic Blood Pressure (SBP), Diastolic Blood Pressure (DBP), Mean Arterial Pressure (MAP), Glasgow Coma Scale (GCS), LOC, O2 Saturation, FiO2, SpO2/FiO2 ratio, SpO2, PaO2, PaCO2, | Continuous |
| **Laboratory values** <br><br><br><br> **(23)** | Basic Metabolic Panel [BMP]: Sodium, Potassium, Bicarbonate (CO2), Anion Gap, Glucose, Calcium, Blood Urea Nitrogen (BUN), Serum Creatinine (SCr) baseline, SCr change, Phosphate | Continuous |
| | Liver Function Test [LFT]: Albumin, Bilirubin baseline, Bilirubin change | |
| | Complete Blood Count [CBC]: White Blood Cells (WBC) and percentage band, Hemoglobin, Platelet Count | |
| | Other labs: D-Dimer, INR, PTT, Fibrinogen, Lactate, pH | |
| **Vital signs and Laboratory value trends** <br><br><br> **(172)** | Initial value of variable vital signs and labs before prediction point[a] | Continuous |
| | Highest value of variable vital signs and labs before prediction point[a] | |
| | Lowest value of variable vital signs and labs before prediction point[a] | |
| | Average value of variable vital signs and labs before prediction point[a] | |
| **Diagnostics present or before admission** <br><br> **(561)** | Infection diagnosis codes (present on admission) (538), comorbidities (16), Charlson comorbidities index, Chronic Conditions (on problem list or medical history) (7) | Binary |
| **Critical Events** <br><br> **(23)** | Identifiers of first occurrence of 2 SIRS (12), first occurrence of distinct sites of organ dysfunctions (7), Triage time of the day (4) | Binary |

[a]prediction point is defined as the time of 3-hour bundle initiation (first occurrence of blood culture order, first antibiotics administration, initial lactate, and fluid bolus) if applicable, or sepsis onset (last occurrent of 2 SIRS, suspected infection and first site of organ dysfunction) if not.

The importance of factors was ranked based on "gain", or the cumulative improvement in AUROC attributed to all splits involving the predictor across all decision trees [20]. The marginal effects were measured by the SHAP (Shapley Additive exPlanations) values [23], which evaluated how the odds ratio changed by including a particular factor of certain value for each individual patient (S1 Appendix). The SHAP value not only captured the global patterns of effects of each factor but demonstrated the patient-level variations of the effects. For each model, we reported the 20 factors that provided the most individual "gain" (i.e. cumulatively accounting for at least 50% "gain"). To interpolate the non-linear factorial effects as well as the uncertainties, we fit cubic splines across with 6 knots over the SHAP values and constructed a bootstrapped confidence interval for each factor [24]. Since the XGBoost implementation of the GBM model incorporated missing value branches for each split of each tree, we were also able to identify if the "missing pattern" of certain factors could have meaningful implications [22].

## Results

The initial cohort contained 25,427 encounters identified as suspected infection, of which 11,590 developed markers of two SIRS and at least one site of organ dysfunction within 48 hours of triage (Fig 1). The mean age of the final cohort was 57 (±17) years, evenly distributed

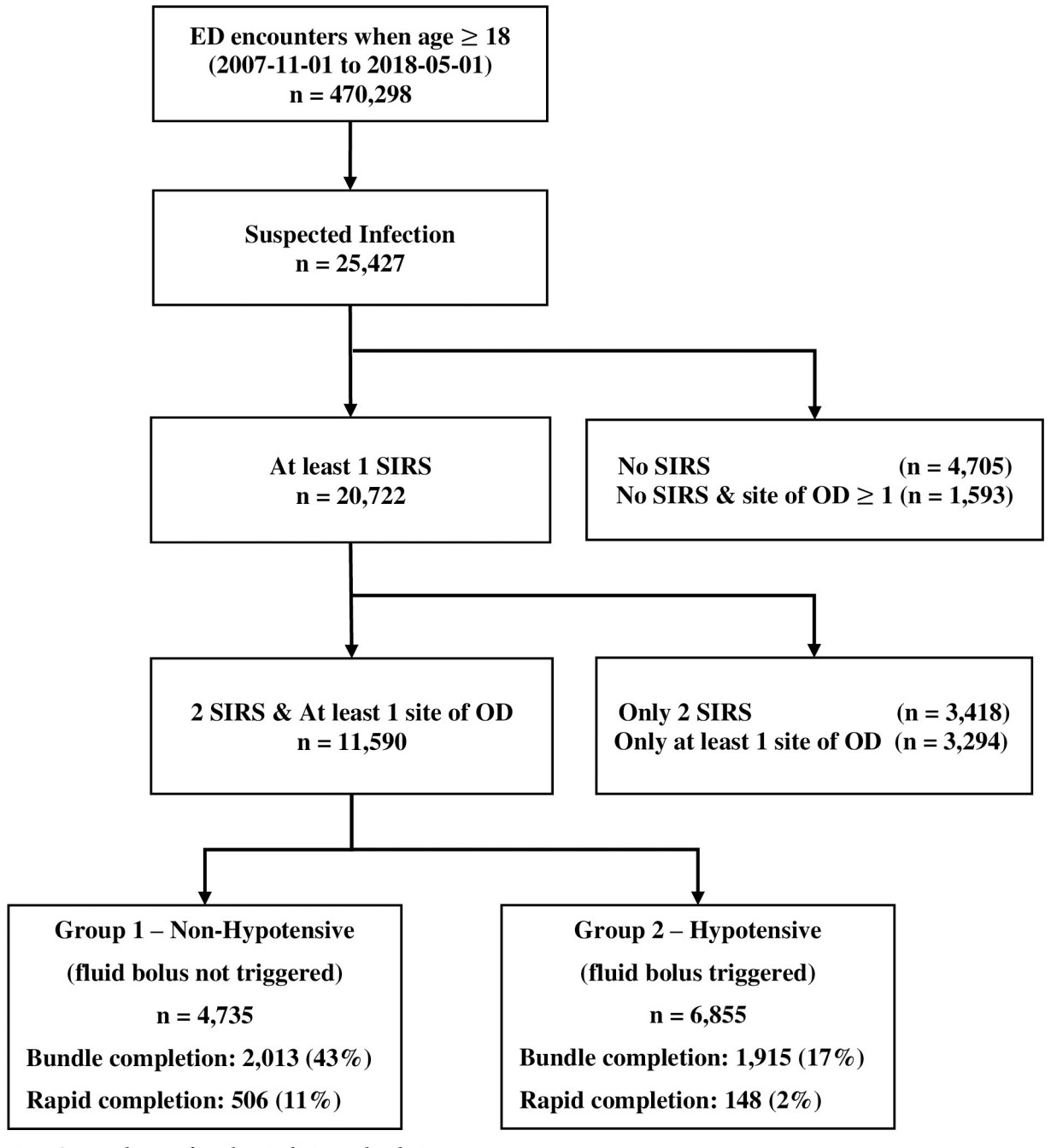

**Fig 1. Consort diagram for cohort inclusion and exclusion.**

between males and females, with the majority being Caucasian. Group 1 included 6,855 (59%) encounters, with 47% women; Group 2 comprised 4,735 (41%) encounters with 55% women. The Logistic Organ Dysfunction Score (LODS) of Group 2 was slightly higher than that of Group 1 (Table 2).

Times of the bundle components with respect to triage are shown in Table 3 and Fig 1. Bundle completion rates were lower (17% vs. 42% for completion, 2.2% vs. 10.7% for rapid completion) and time for completion of the bundle longer (27.5 [IQR: 10.4–80.0] hours vs 4.7

**Table 2. Demographic and physiological characteristics.**

| Demographic Characteristic | | Overall | Group 1 | Group 2 |
|---|---|---|---|---|
| | | | (n = 4,735) | (n = 6,855) |
| Age, *mean (sd)* | | 57 (17) | 57 (16) | 58 (18) |
| Sex, *n (%)* | | | | |
| | Female | 6,068 (52) | 2,202 (47) | 3,773 (55) |
| | Male | 5,522 (48) | 2,533 (53) | 3,062 (45) |
| Race, *n (%)* | | | | |
| | White | 7,633 (66) | 2,917 (62) | 4,573 (67) |
| | Black | 2,570 (22) | 1,278 (27) | 1,367 (20) |
| | Asian | 166 (1) | 52 (1.1) | 109 (1.6) |
| | Other[a] | 1,223 (11) | 469 (9.9) | 787 (11.4) |
| Ethnicity, *n (%)* | | | | |
| | Non-Hispanic | 10,640 (92) | 4,385 (93) | 6,281 (92) |
| | Hispanic | 918 (8) | 350 (7) | 574 (8) |
| **Physiological Characteristics** | | | | |
| Initial Temperature (˚C) | *mean (sd)* | 37.3 (1.20) | 37.3 (1.15) | 37.3 (1.22) |
| Initial Hear Rate (/min) | *mean (sd)* | 108 (19.7) | 108 (17.8) | 108 (20.4) |
| Initial Respiratory Rate (/min) | *mean (sd)* | 22 (6.7) | 22 (6.2) | 22 (7.0) |
| Initial WBC counts (K/uL) | *median (IQR)* | 14 (9.4, 19) | 13 (8.6, 17.9) | 14 (9.7, 19.7) |
| Temperature ≥38˚C or ≤36˚C[b] | *n (%)* | 5,126 (44) | 1,772 (37) | 3,213 (47) |
| Heart Rate ≥90/min[b] | *n (%)* | 10,792 (93) | 4,403 (93) | 6,386 (93) |
| Respiratory Rate ≥20/min[b] | *n (%)* | 9,518 (82) | 3,731 (79) | 5,718 (83) |
| WBC counts >12K/uL or <4K/uL | *n (%)* | 9,623 (83) | 3,782 (80) | 5,776 (84) |
| **Logistic Organ Dysfunction Score** | | | | |
| LODS within first 3hr since triage | *mean (sd)* | 1.9 (1.80) | 1.9 (1.59) | 2.5 (2.12) |
| LODS prior to initial antibiotics | *mean (sd)* | 2.5 (2.05) | 2.1 (1.62) | 2.6 (2.17) |

[a]The catch-all "Other" category includes: American Ind/Pac Islander/Two Races, Other and Unknown.

[b]All the SIRS events are captured within first 48 hours during ED stay.

[IQR: 3.3–8.0] hours from triage), in Group 1 than in Group 2. Low bundle completion was principally associated with prolonged time to IV fluid completion; 57% of Group 2 patients completed the other three bundle components (blood culture, antibiotics administration, and initial lactate) within 3 hours after triage.

Model 1 was built on Group 1 patients, and ultimately selected 142 discriminant factors. Model 2 was developed for Group 2 patients and selected 158 important factors. As shown in Fig 2, both models showed good predictive ability for rapid completion of sepsis bundles based on AUROC in the validation cohort; 0.84 [95% CI, 0.81–0.86] for Model 1 and 0.91 [95% CI, 0.86–0.95] for Model 2. The optimal sensitivity and specificity for Model 1 were 81% [95% CI, 72% - 87%] and 74% [95% CI, 70% - 83%], and were 91% [95% CI, 81% - 97%] and 83% [95% CI, 79% - 87%] for Model 2. At the points of optimal sensitivity, Model 1 achieved a PPV of 40% [95% CI, 36% - 42%], and Model 2 achieved a PPV of 44% [95% CI, 40% - 47%]. Both models achieved competitive AUPRCs (0.41 [95% CI, 0.37–0.43] for Model 1 and 0.29 [95% CI, 0.20–0.41] for Model 2) in comparison to the baseline rates of 10.7% and 2.2%. Both models show good calibrations with p-value > 0.1 for the HL test. The Lift Curves for both models suggest good discriminative power as the higher the risk decile, the more rapid treatment cases the risk decile includes. In addition, the model performance was consistent across calendar years (Fig 3).

**Table 3. Bundle component timing.**

| Bundle Components | Group 1 (n = 4,735) | Group 2 (n = 6,855) |
|---|---|---|
| | n (%, median hours since triage [IQR]) | n (%, median hours since triage [IQR]) |
| Blood Culture | 3,923 (83%, 1.3 [0.8, 4.3]) | 5,857 (85%, 1.3 [0.6, 4.5]) |
| Antibiotic Administration | 4,032 (85%, 3.2 [2.1, 5.4]) | 6,098 (89%, 2.9 [1.7, 4.6]) |
| Initial Lactate | 2,533 (53%, 2.9 [2.1, 5.8]) | 4,387 (64%, 3.4 [2.7, 6.0]) |
| IV Fluid Bolus Begin | NA | 5,402 (79%, 7.4 [2.7, 27.4]) |
| IV Fluid Bolus Complete | NA | 3,150 (46%, 28.4 [10.2, 79.2]) |
| **Bundle Completeness** | | |
| At least initiated | 4,573 (97%, 1.2 [0.4, 3.4]) | 6,855 (100%, 0.9 [0.4, 2.4]) |
| At least initiated therapeutic components | 2,013 (42%, 4.2 [3.1, 6.4]) | 3,068 (26%, 3.6 [2.3, 5.2]) |
| Completion of bundle components (except for IV Fluid Bolus) | 1,393 (29%, 4.7 [3.3, 8.0]) | 3,920 (57%, 4.4 [3.0, 7.2]) |
| Completion of all bundle components | 1,393 (29%, 4.7 [3.3, 8.0]) | 1,925 (17%, 27.5 [10.4, 80.0]) |
| Rapid completion of bundle components | 506 (10.7%, 2.0 [1.4, 2.6]) | 148 (2.2%, 1.8 [1.3, 2.3], 1.0 [0.4, 1.4][a]) |

[a]The first time is the completion of bundle components since triage (except for IV fluid bolus); the second time is the completion of 2 L bolus since fluid initiation.

The middle point of each bar represents the corresponding performance metric over validation set within certain calendar year group. Upper and lower bounds of each bar correspond to 95% bootstrapping confidence interval for each metric. The model performance was consistent across calendar years.

A Spearman correlation test (0.6 [0.43–0.76]) suggested that the feature rankings of the two models were statistically different. However, both models identified 8 common risk factors among the top 20 factors: maximum Glasgow Coma Scale (GCS), minimum heart rate, initial temperature, initial WBC count, initial serum creatinine (SCr), minimum diastolic blood pressure (DBP), initial platelet count and age (Fig 4). Among Group 1 patients, the majority of the top 20 most impactful factors to the model were components of the initial physiological profiles, such as: increased bilirubin on first measure or bilirubin increase from pre-hospitalization baseline, arterial pH, blood pressure, heart rate, respiratory rate, SpO$_2$, and INR. The most impactful factors specific to Group 2 patients were more likely to represent the minimum, maximum, or mean of values before the prediction point: minimum and maximum mean arterial pressure, minimum and maximum systolic blood pressure (SBP), mean heart rate, and maxima of respiratory rate, temperature, or total CO$_2$.

Fig 5 further depicts the full details of marginal effects of top 12 most impactful features for both models (for better resolution, we reported the remaining marginal effects for next top 13–20 most impactful features in S2 Appendix), which can be used to identify specific value ranges that have strong positive or negative association with rapid treatment. For both Group 1 and Group 2 patients, the presence of a recorded GCS was associated with a reduction in odds ratio of rapid treatment by a factor of 0.3 to 0.6, while the absence of a recorded GCS was associated with an increase of odds ratio by a factor of 1.2 to 1.3. A more rapid minimum heart rate was associated with increasing likelihood of rapid treatment. The odds ratio for rapid treatment was significantly increased among both Group 1 and Group 2 patients when the minimum heart rate was ≥100 beats/min. Initial temperature showed a U-shaped relationship,

| Prediction Performance | Model 1 | Model 2 |
|---|---|---|
| AUROC, [95% CI] | 0.84, [0.81, 0.86] | 0.91, [0.86, 0.95] |
| AUPRC, [95% CI] | 0.41, [0.37, 0.43] | 0.29, [0.15, 0.41] |
| Optimal Sensitivity, [95% CI] | 0.81, [0.72, 0.87] | 0.91, [0.81, 0.97] |
| Optimal Specificity, [95% CI] | 0.74, [0.70, 0.83] | 0.83, [0.79, 0.87] |
| Optimal PPV, [95% CI] | 0.40, [0.37, 0.42] | 0.34, [0.20, 0.47] |
| Brier Score, [95% CI] | 0.08, [0.08, 0.09] | 0.07, [0.06, 0.08] |
| Calibration plot<br><br>(dashed line the 45º reference line) |  | |
| Hosmer-Lemeshow score, P-value | 13.45, 0.10 | 12.93, 0.15 |
| Lift Curve<br><br>(dashed line shows the base rate of rapid treatment for each model, 1 to 10 represent bins with increasing predicted probabilities of rapid treatment) |  | |

Fig 2. Prediction performance metrics.

with significant increase of odds ratio when $\leq 36°C$ or $\geq 39°C$ for the non-hypotensive patients. However, the relationship of temperature to rapid treatment was more monotonic among Group 2 patients, with significant increases in odds ratio for rapid treatment when initial temperature was $\geq 39°C$. The odds ratio for rapid treatment was increased by a factor of 1.5 by an initial SCr $\geq 1.5$ mg/dL among Group 2 patients; however, the magnitude of this effect was substantially lower and less consistent among Group 1 patients.

Other features associated with significant increases in odds ratio for rapid treatment were specific to either group. For Group 1 patients (Fig 5A), an initial bilirubin increase from baseline by $\geq 1.0$ mg/dL, initial arterial pH value $\leq 7.4$, initial WBC count $\geq 20,000/mm^3$, or an initial heart rate $\geq 120$ beats/min were associated with significant increased odds ratios for rapid treatment. For Group 2 patients (Fig 5B), multiple representations of blood-pressure-related factors were shown to be important, such as minimum SBP, DBP and MAP, maximum

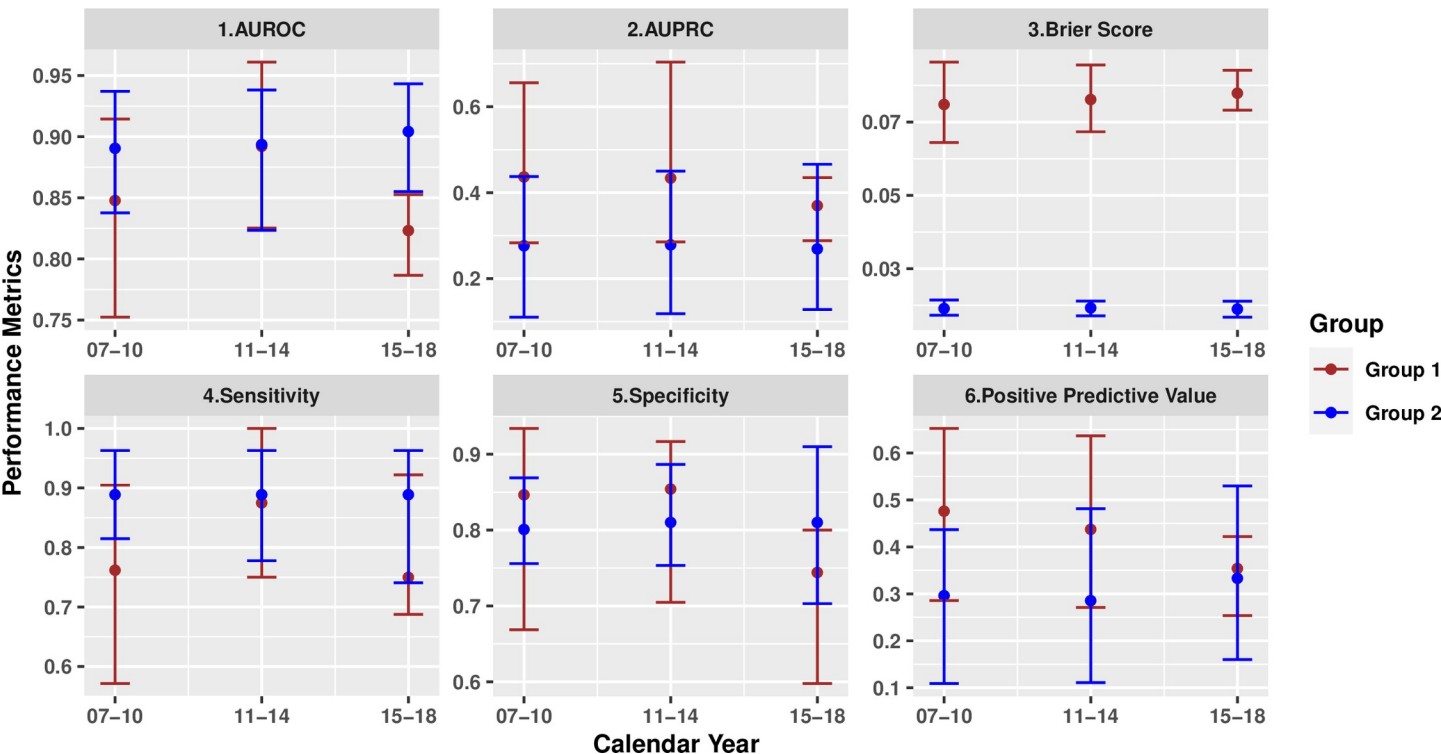

**Fig 3. Model performance comparisons over calendar years.**

SBP and MAP, and mean SBP. However, these values were not necessarily correlated with one another (Fig 6). A minimum SBP or MAP $\leq 90$ mmHg, or a mean SBP $\leq 100$ mmHG was significantly associated with increased likelihood of rapid treatment, while a minimum DBP $\geq 60$ mmHg was associated with moderately higher chance of rapid treatment. Additionally, a mean heart rate $\geq 120$/min, and/or an initial SCr increase $\geq 1$ mg/dL, and/or missing maximum total $CO_2$ or temperature were associated with significantly increased odds ratios for rapid treatment (S2 Appendix).

## Discussion

Early and aggressive treatment of sepsis with antibiotics and fluids can be lifesaving [25–28]. Yet, in spite of educational efforts, reporting measures, and even regulations mandating hospital protocols for sepsis diagnosis and treatment, many patients who should be rapidly treated are not [2, 29, 30]. An understanding of what patient factors are associated with rapid treatment (or slow treatment) may allow for beneficial changes in education, individualized presentation of data in the EMR, or consistency in approach to septic patients.

We used a data-driven machine learning approach to evaluate which clinical features that are present early in a patient's hospital course have been associated with rapid sepsis treatment by physicians in our institution. Our initial investigations began with logistic regression modeling. However, this modeling was limited in two important ways. First, the variables to input into the model were based upon prior clinical knowledge and did not allow for discovery of latent variables that may be important but that were not already intuitive. Second, logistic regression forces linear and independent behavior of variables that, themselves, may be non-linear and highly correlated.

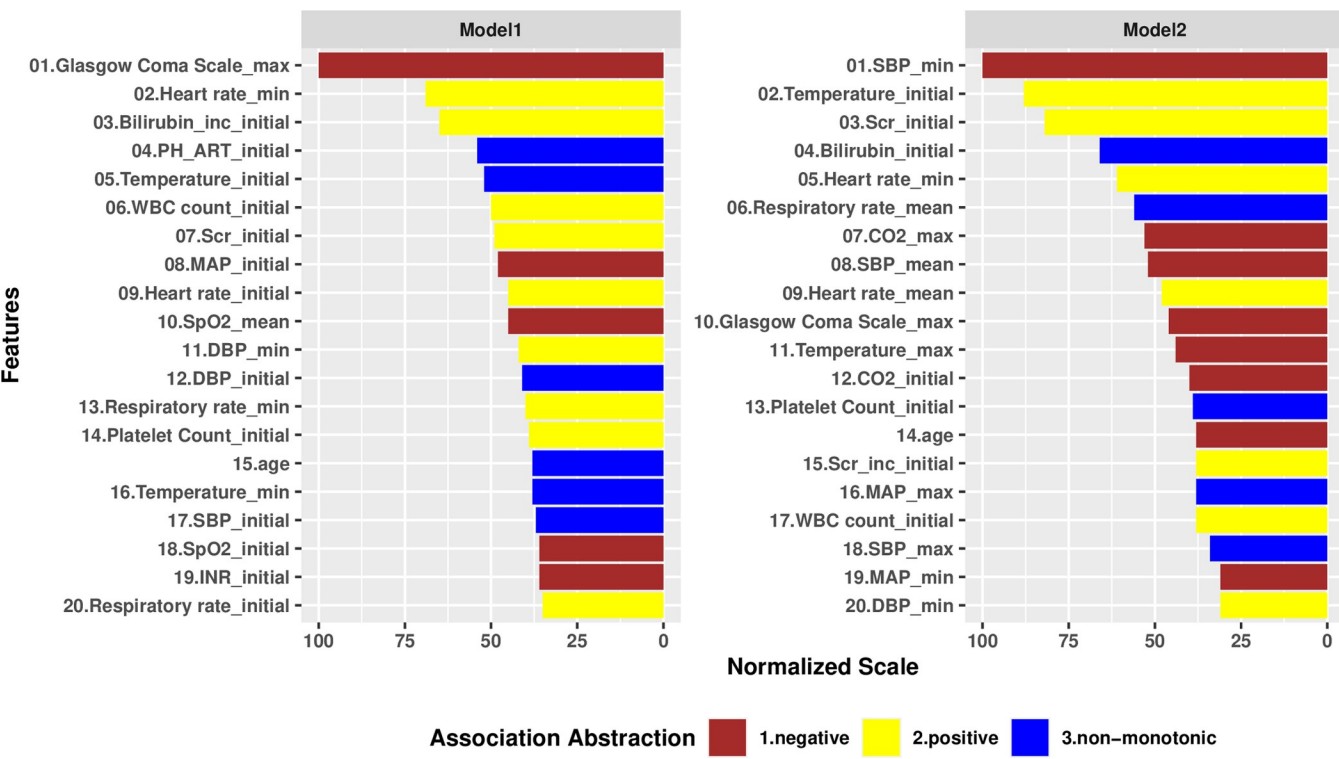

**Fig 4. Variable importance plot for Model 1 and Model 2.** The importance score of each variable has been scaled to a maximum value of 100. The colors indicate marginal associations of variables with rapid treatment, which are abstractions calculated by comparing SHAP values at 25th, 50th and 75th percentiles of the variable values.

Our study spanned more than a decade of care that encompassed both Sepsis-2 and Sepsis-3 diagnostic criteria, as well as continuous quality improvement efforts to speed care delivery for sepsis. Because the elements of the treatment bundle are relatively basic—antibiotics and intravenous fluids—it is likely that the specific bundle recommendation and the timing recommended by the bundle play only a small role in the rapidity with which physicians treat septic patients. We believe that we have identified patient features that may well promote rapid delivery of basic resuscitative therapies, regardless of the timing recommendations by the SSC guidelines. We developed two models for subgroups, determined by the presence or absence of hypotension or lactate ≥ 4. Both models demonstrated good performance. Model 2 demonstrated better predictive ability, indicating that rapid delivery of treatment bundles is more uniform in these patients.

Although each model identifies more than 100 factors associated with rapid treatment, some general patterns based on the most impactful clinical features emerge. For example, instead of specific values of the SIRS criteria or counts of SIRS criteria met, we identified that ranges of values for temperature, WBC count, heart rate and respiratory rate were associated with rapid 3-hour bundle treatment. This suggests that physicians rely on ranges of findings, rather than on specific thresholds when making treatment decisions. Some findings were intuitive and predictable, such as that minimal blood pressure among hypotensive patients was associated with more rapid treatment. Other findings are less intuitive, such as that the presence of any recorded GCS was associated with slower treatment. GCS was less likely to be frequently recorded when it was normal. We hypothesize that patients with abnormal GCS may slow sepsis treatment, because they first receive CT scans or other evaluations to evaluate other

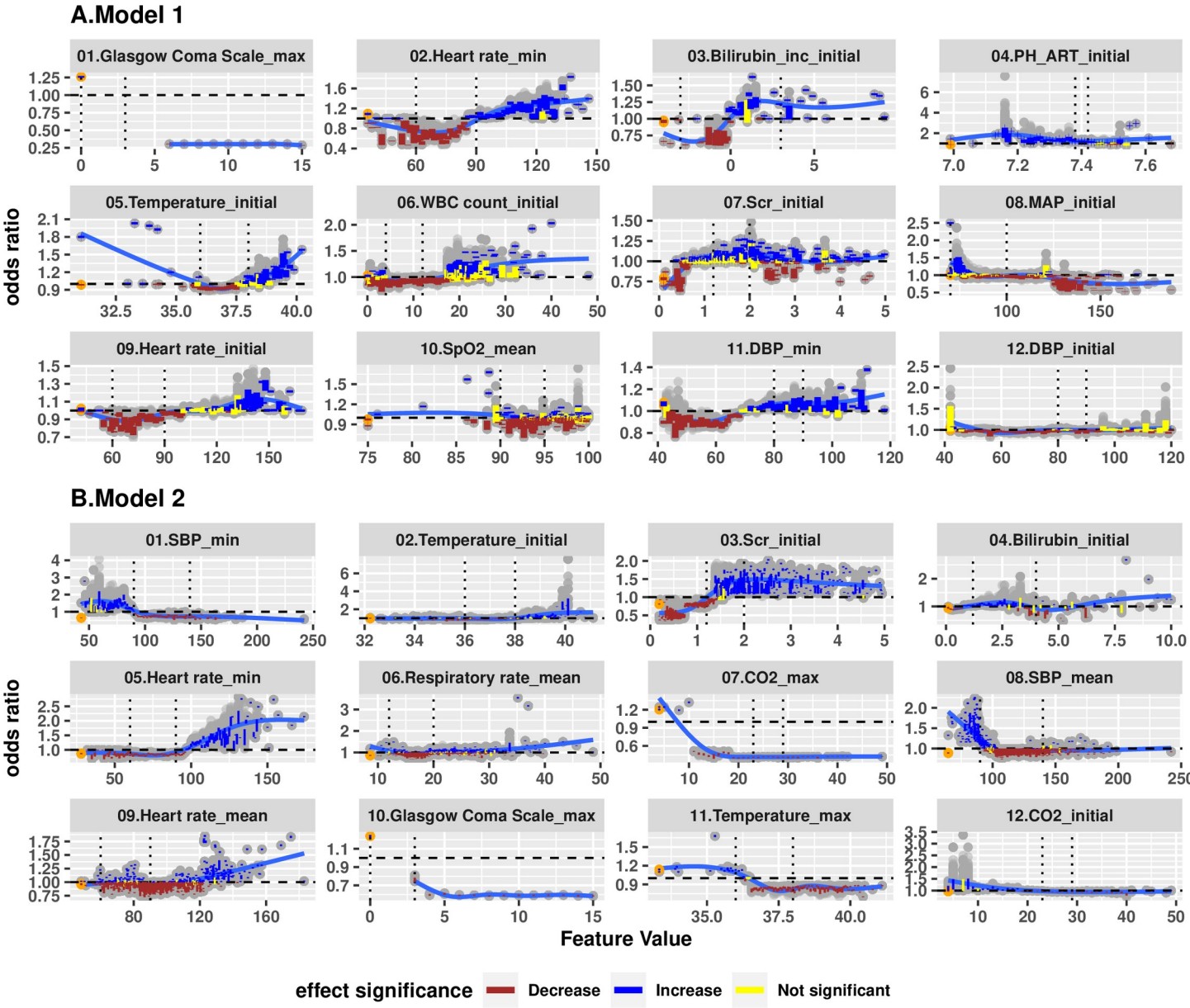

**Fig 5.** Marginal effects of variables ranked top 12 for Model 1 (Panel A) and Model 2 (Panel B) based on SHAP values, i.e. exponential of the SHAP value. Each dot represents an average change of odds ratio for a variable, taking certain values within a bootstrapped sample. Each colored vertical line depicts a 95% bootstrap confidence interval based on 100 bootstrapped samples. A brown line suggests an odds ratio change significantly higher than 1.0; a blue line suggests an odds ratio change significantly lower than 1.0; a yellow line suggests an odds ratio not significantly different from 1.0. Orange dots represent the odds ratio effect of not having the particular data point recorded for the model. The dashed horizontal line shows an odds ratio of 1.

causes of abnormal mental status. Age is also an important feature for rapidity of treatment, such that patients over 60 years of age who were not hypotensive received treatment more rapidly than younger patients. However, among hypotensive patients age over 70 was associated with slower treatment (S2 Appendix).

Rather than focusing on features that predict the diagnosis of sepsis, these studies focus on features that predict more rapid treatment of sepsis. To the extent that more rapid treatment is associated with improved sepsis outcomes, the data could be of use in designing more effective

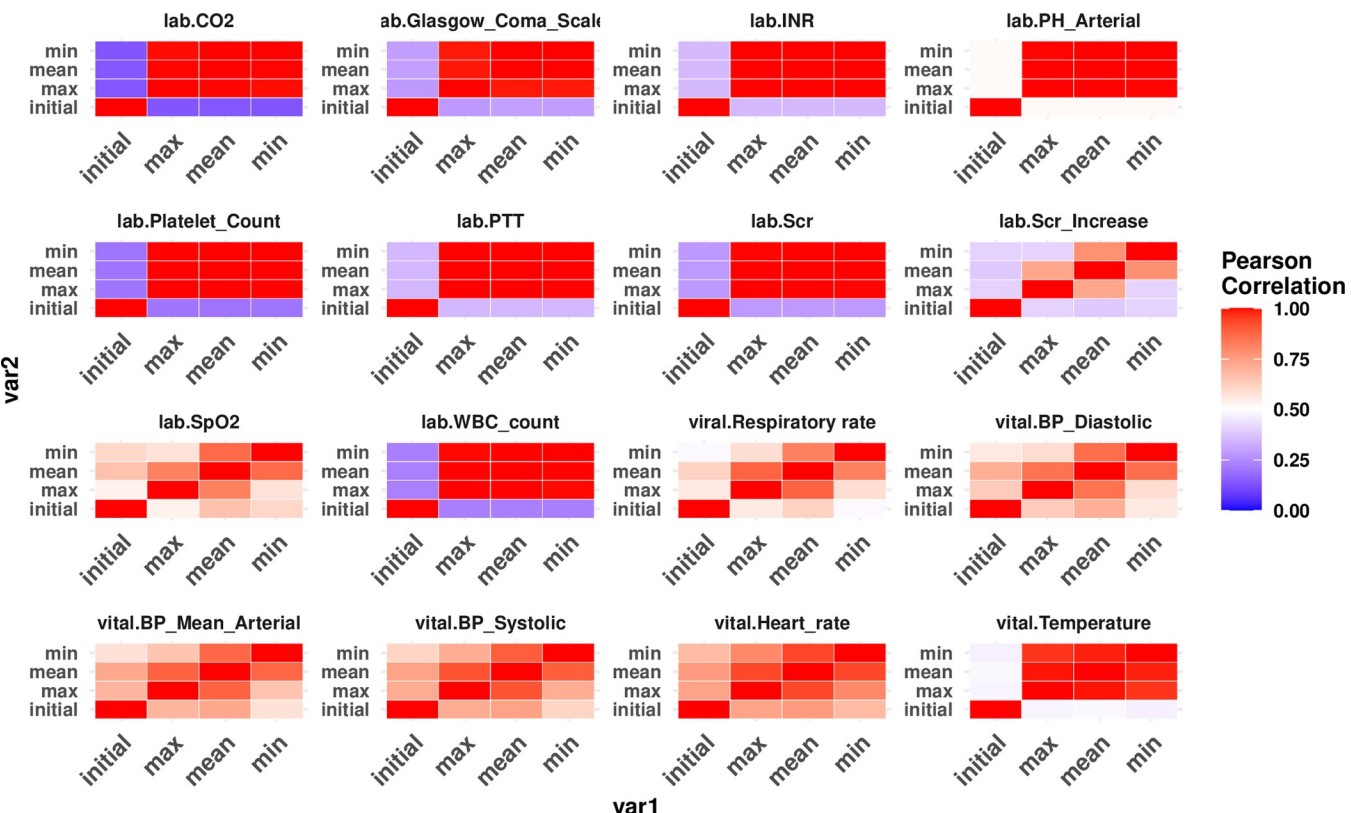

**Fig 6. The correlation heatmap among different abstractions of the same clinical variable with repeated measurement.** Note that the "Initial" values are not always very different from the other types of summaries.

EMR presentations that provide physicians with improved situational awareness and promote desired behavior. These studies cannot establish causality, but they may provide insights into factors that motivate physicians to treat faster when sepsis is present. Much as Amazon promotes specific products or Google promotes specific advertisements based on modeling of previous behavior, alerting mechanisms that highlight the appropriate features when they are present, i.e. those features historically associated with more rapid treatment, could more effectively stimulate providers to speed therapy than do simple alerts based on the presence or absence of sepsis. Because physician behavior varies from one hospital to another, incorporation of these findings in the EMR would require local model implementation and continuous learning within each hospital, which could be accomplished either by local servers or by a cloud implementation that could be informed by data from many hospitals. As discussed in [31], awareness and compliance with protocols can be difficult to maintain in the absence of an effective clinical pathway, which requires careful planning and dedicated resources. A better understanding of alerting features associated with rapid treatment would provide useful and evidence-based insights to a better design for creating such clinical pathways and continuous education.

The majority of the most impactful patient features involve vital signs data. Although that information might have been anecdotally predicted, we believe this to be the first study demonstrating associations of vital signs data with rapidity of treatment for sepsis. Certain features are machine generated, such as mean respiratory rate over time, which is not a known diagnostic or prognostic parameter in sepsis. However, it seems possible, even likely that

physicians may, in fact, consciously or unconsciously make judgements and take actions related to it. Though it would be of interest, it is not necessary to understand exactly why or how the clinical features promote the desired behavior, which is to say that our studies do not predict the physician thought process, only the outcome of that process.

Our study has several limitations. Since we used only structured EMR data, the definition of suspected infection was based on clinicians' perceptions of infection, as evidenced by the actions of obtaining blood cultures and initiating antibiotics; we are unable to ascertain more detail about how and why infection was suspected. The actual indication that a physician has diagnosed infection is routinely captured in free-text clinical notes; to further this research, one would need to incorporate free-text nursing and physician notes using natural language processing tools. The diagnostic criteria we used were those of the Sepsis-1 and 2 consensus conferences, chosen because they were the extant diagnostic criteria throughout the vast majority of the study period, and physicians and other providers would have been familiar with them. Lactate value was not used in the learning even though it may be a strong predictor, because of the need to prevent "label leakage", since the prediction targets for both models under the Sepsis 1 and 2 criteria are partially defined by initial lactate. This factor would be mitigated in Model 1, if Sepsis 3 diagnostic criteria were applied to the cohort, since lactate is not a defining feature of sepsis in those criteria. Our data sources do not include social factors that may impact rapidity of treatment, such as presenting symptoms [32], income level, educational level, occupation, or zip code, nor were we able to capture such features as staffing ratios, pharmacy volumes, or ER wait times; these factors could well affect clinician recognition of sepsis and treatment decision making. We did find that time of day did not significantly affect rapid sepsis treatment (Table 1). Data are analyzed on an encounter level, and it is possible that a given patient could exist in both the training data and the validation data. Finally, we have demonstrated association, not causation, and it is impossible to determine whether we have described overt patient characteristics or latent provider characteristics. However, we have no record of individual provider characteristics that could inform the models. Further, the data provide no insights into how providers arrived at decisions to treat patients with the associated features, only that they did, i.e. the models do not replicate physician thought processes in any way, but are predictive of physician behavior.

## Conclusion

We developed machine-learning models for accurately predicting rapid treatment of patients with sepsis in the emergency department and identified clinical factors that are commonly available and that physicians may recognize and use but that are not a part of standard thresholds. These studies may be useful to inform a new generation of EMR sepsis alerting tools.

## Supporting information

**S1 Appendix. Examples of SHAP value representations.**
(PDF)

**S2 Appendix.** Marginal effects of variables ranked top 11–20 for Model 1 (Panel A) and Model 2 (Panel B) based on SHAP values, i.e. exponential of the SHAP value.
(PDF)

## Author Contributions

**Conceptualization:** Anurag Patel, Steven Q. Simpson.

**Data curation:** Xing Song.

**Formal analysis:** Xing Song.

**Funding acquisition:** Lemuel R. Waitman, Anurag Patel, Steven Q. Simpson.

**Investigation:** Xing Song, Steven Q. Simpson.

**Methodology:** Xing Song.

**Project administration:** Steven Q. Simpson.

**Resources:** Lemuel R. Waitman.

**Supervision:** Steven Q. Simpson.

**Validation:** Xing Song.

**Visualization:** Xing Song.

**Writing – original draft:** Xing Song.

**Writing – review & editing:** Mei Liu, Lemuel R. Waitman, Anurag Patel, Steven Q. Simpson.

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
