## [Decision Letter · Decision Letter 0]

28 Sep 2020

PONE-D-20-06811

Clinical Factors Associated with Rapid Treatment of Sepsis

PLOS ONE

Dear Dr. Song,

Thank you for submitting your manuscript to PLOS ONE. After careful consideration, we feel that it has merit but does not fully meet PLOS ONE’s publication criteria as it currently stands. Therefore, we invite you to submit a revised version of the manuscript that addresses the points raised during the review process.

We look forward to receiving your revised manuscript.

Kind regards,

Robert Moskovitch

Academic Editor

PLOS ONE

2.We note that you have indicated that data from this study are available upon request. PLOS only allows data to be available upon request if there are legal or ethical restrictions on sharing data publicly. For information on unacceptable data access restrictions, please see http://journals.plos.org/plosone/s/data-availability#loc-unacceptable-data-access-restrictions.

3.Thank you for stating the following in the Financial Disclosure section:

[SQS and PA received Blue KC Outcome Research Grants (No.0925-0001) and the authors played role in study design, decision to publish and preparation of the manuscript.

LR received CTSA grant UL1TR002366 from NCRR/NIH and the author played role in data collection and preparation of the manuscript.].   

We note that one or more of the authors are employed by a commercial company: Anurag4Health

Additional Editor Comments (if provided):

Please refer carefully to the reviewers' comments.

Reviewers' comments:

Reviewer's Responses to Questions

**Comments to the Author**

1. Is the manuscript technically sound, and do the data support the conclusions?

Reviewer #1: Partly

Reviewer #2: Yes

Reviewer #3: Yes

2. Has the statistical analysis been performed appropriately and rigorously? 

Reviewer #1: Yes

Reviewer #2: Yes

Reviewer #3: Yes

3. Have the authors made all data underlying the findings in their manuscript fully available?

Reviewer #1: No

Reviewer #2: No

Reviewer #3: No

4. Is the manuscript presented in an intelligible fashion and written in standard English?

Reviewer #1: Yes

Reviewer #2: Yes

Reviewer #3: Yes

5. Review Comments to the Author

Reviewer #1: High-level comments:

The paper is relatively clearly written, and of mild interest, although its impact and significance are somewhat limited.

The paper discusses a data analysis technique, and model analysis technique (SHAP) applied to a concrete medical problem (rapid treatment of Sepsis).

The topic of Sepsis is a bit specific, but the methodology can be used for different types of complications.

The results and the analysis that are presented in this paper have some interest in the medical domain, but it is not a substantial contribution.

The contributions have to be carefully state and rewritten.

I thought it was unfortunate that the authors chose not to submit supplementary material that could have included additional details about the XGBoost parameters.

Low-level comments:

Abstract - "ED" - was used before it was defined. You explained it on page 4.

Abstract - "These machine learning methods identified factors associated with rapid treatment of severe sepsis patients from a large volume of high-dimensional clinical data" - What about the combination of these factors? The combination of these factors leads to Sepsis and not separately.

Page 3 - "Sepsis is an important public health problem in the United States and is the leading cause of death among hospitalized patients" - Where is the definition of Sepsis? Please define Sepsis clearly.

Page 3 - SIRS - again, was used before it was defined. You explained it on Page 4.

Any similar research with SHAP and Sepsis or other complications? Write more about it in the introduction.

The PDF contains low-resolution figures; it's hard to understand, especially when printing. For example, I can't see the features in Figure 4.

Page 5 - 'We randomly partitioned the data' - why randomly? Try with stratified k-fold cross-validation (same class distribution).

Page 6 - "To control overfitting, we carefully tuned the model hyper-parameters within the training set using 10-fold cross-validation." - Why not training, validation, and testing set? You can use the validation set for parameter tuning.

Your explanation about the missing values is not clear to me.

You mentioned SHAP on page 6 but explained it only on page 7.

It is not clear how do you treat the continuous variable. Aggregation over time? It will be better to add figures to explain how exactly you generate the features over time.

Why GBM? Justify your selection. You explained why not logistic regression. Can you tell why not, for example, Random forest? Neural network?

"logistic regression forces linear and independent behavior of variables that, themselves, may be non-linear and highly correlated." And GBMs? Please explain why it's different.

Page 7-8 - "Since the XGBoost implementation of the GBM model incorporated missing value branches for each split of each tree, we were also able to identify if the "missing pattern" of certain factors could have meaningful implications (14)." - Is XGBoost the only one?

Why you put the demographic table on the results section?

Add the research questions before the results.

"These studies cannot establish causality, but they may provide insights into factors that motivate physicians to treat faster when sepsis is present." - I think it's important to emphasize this in the introduction.

Please share your insights in the Conclusions section.

"Because physician behavior varies from one hospital to another, incorporation of these findings in the EMR would require local model implementation and continuous learning within each hospital, which could be accomplished either by local servers or by a cloud implementation that could be informed by data from many hospitals." - This sentence is hard to understand. Consider also trying your method on public data, such as MIMIC III. Any additional information you can add for readers that want to replicate your experiments? Assuming other researchers hold your data.

Did you use the same seed for both models (for group 1, 2)?

Minor points that could be usefully clarified:

Line numbering is part of the template. Did you use another template?

Please prepare figures that are readable in B&W (I think it already made for colorblind).

Reviewer #2: The paper defines an interesting research question, and addresses it using Gradient boosting models on ED visits data. Overall, the manuscript is well-written; however, there are several shortfalls that need to be addressed.

Please refer to the attached file for the detailed comments.

Reviewer #3: In the manuscript titled "Clinical Factors Associated with Rapid Treatment of Sepsis" by Song et. al. the authors described how they used data from University of Kansas Medical Center. They used retrospective data of 10 years period for finding clinical factors indicative for rapid treatment of features.

Limitation of current systems is that EMR is static and present only data that was predefined to be shown for specific use. Although other data is available, it may be buried under menus and submenus.

The uniqueness of the current study is that rather than focusing on features that predict the diagnosis of sepsis, this study focus on features that predict more rapid treatment of sepsis.

Overall, the manuscript is well written and the analysis was done adequately. I do have a few comments that can be used to improve the manuscript. The definitions of inclusion exclusion criteria of for defining cases/controls and for thresholding were not clear. See bellow related comments.

The next sentence in the Introduction is not clear: "To further exclude patients who were admitted through the ED but developed sepsis later in their hospitalization, we inferred an hour boundary based on the timing distribution of patients with infection present on admission, which was 13 hours since triage." What is this hour boundary, and when is it on the timeline of the admission, and for what proposes?

Sepsis is also a critical issues in ICUs. The authors did not mention that point. Please explain whether the model could be applicable to ICUs. If possible, test the model in the same medical center in the ICU unit.

A reference so SSC document is missing on page 5.

"EMS (personnel)" is used without definition of EMS abbreviation.

Definition of the two groups is not clear. What is the exact differences between the two groups and what is shared, and why was it defined as such? Is it based on previous studies, or clinical guidelines or something else?

GBM was trained on 70% of patients. What if a patient was admitted more than once to the ED?

In the discussion, authors state that "Data are analyzed on an encounter level, and it is possible that a given patient could exist in both the training data and the validation data." This is not how it was presented in the introduction where the counts represented number of patients. This is critical limitation of a patient can appear both in training and validation as sepsis is a risk for future sepsis. Instead, authors should eliminate multiple samples of the same patient (by random, or by chronological order), or leave all admission of same patient in the same split (training or validation).

Whatsoever, this decision cannot be buried in the discussion but must be explicitly stated in the methods.

"At each iteration during the training stage, we sampled positive and negative cases in equal proportion" Does equal proportion mean 1:1, or proportional to the overall proportion of positive and negatives, or the proportion of each node?

Moreover, what are positive and negative cases? I assume it's relate to timely completion or rapid completion, but author need to be explicit, as it was defined well above.

"(0, 1, 2, 3, NA, and NA)" shouldn't it be "(0, 1, 2, 3, and NA)"?

Please state version of R used and version of the other packages xgboost and xgboostExplainer.

Table 1:

- No need for "Basic Metabolic Panel [BMP]:", "Liver Function Test [LFT]: " as the details are followed.

I guess the numbers in parentheses is the number of features. But the details do no match. For example, there are no 20 listed vital signs.

List of diagnosis codes or variables should be given in the supplement or as a reference where it is listed.

How come "Triage time of the day" gives 4 features? Time of the day is a single datum.

Figure 1 doesn't match numbers in text. For example in the text, Group 1 comprised of 6855 patients, while in Figure 1 it is group 2 that have this number of samples.

Figures' resolution are too low, and have to be improved.

The authors claimed based on Figure 3 that "model performance was consistent across calendar years". First, should it be 'models'?

Second, such claim needs to be supported statistically.

"A Spearman correlation test (0.6 [0.43 - 0.76]) suggested that the feature rankings of the two models were statistically different." What is 0.6? Is it the r squared, or test's p-value? A non-significant p-value does not implies that models are statistically different.

How many features were used for this test? All of them or top 20?

Authors need to compare not only the top features but their directionality.

When comparing model 1 and model 2's features the authors claim that among top factors the initial physiological were present for M1, while for M2 there were more likely to represent values before prediction point. The author need to quantify it.

It is not clear to me how author see a reduction in OR of absent of GCS. Is it reflected in figures 5,6?

Figure 7 is not discussed, and therefore should be eliminated.

The authors used one method for two purposes. 1. Outcome prediction. 2. Identify important features. Although this two objectives are related, they do not have to be coupled. So there are methods that are better for predictions and other methods for finding top affecting features.

Data was not made available.

Code was not shared. Although this is not a requirement, but it became acceptable that authors publish their code in an open to the public repository as a package or as a notebook.

6. PLOS authors have the option to publish the peer review history of their article (what does this mean?). If published, this will include your full peer review and any attached files.

Reviewer #1: No

Reviewer #2: No

Reviewer #3: **Yes: **Nadav Rappoport

---

## [Author Response · Author response to Decision Letter 0]

8 Dec 2020

The manuscript applies association rule mining to identify features that are associated with rapid treatment of Spesis patients in Emergency Department (ED). The paper defines an interesting research question, and addresses it using Gradient boosting models on ED visits data. Overall, the manuscript is well-written; however, there are several shortfalls that need to be addressed:

Assumptions and Scope:

Authors mentioned several key assumptions and definitions throughout the paper. As a reader, I would like to see them all at one place, preferably toward the beginning of the paper as bullet-points. Currently, they are defined as needed all over the paper. For example, “Since sepsis, so defined, was considered to be present on admission…”in page 5. Also, “…components of the 3-hour bundle represent standard elements of excellent sepsis care and have always been present in well-treated patients” in page 5. Similarly, “We hypothesize that patients with abnormal GCS may slow sepsis treatment, because they first receive …” in page 14. To set the stage, most of the key limitations in Page 15 (e.g. “definition of suspected infection was based on clinicians’ perceptions of infection”) need to be mentioned earlier, so the readers exactly know what the scopes are.

Response: We thank reviewer for the suggestions on reorganizing the paper to achieve better clarity. We have moved the key assumptions and definitions toward the beginning of the paper on page 4 and 5. More specifically, 

- We have defined the study cohort using bullet points on page 5.

- the “Since sepsis, so defined, was considered to be present on admission…”on page 5, is now integrated in the exclusion criteria: “To further exclude patients who were admitted through the ED but developed sepsis later in their hospitalization, i.e. to include only sepsis present on admission, we inferred an hour boundary based on the timing distribution of patients with infection present on admission, which was 13 hours since triage.” on page 5.

- the “…components of the 3-hour bundle represent standard elements of excellent sepsis care and have always been present in well-treated patients” in page 5, is now integrated in: “The outcome of interest was timely completion of the Surviving Sepsis Guidelines (SSC) 3-hour bundle components (10). We chose the SSC bundles not as an endorsement, but as quantifiable, time-stamped, and recorded actions that are representative of rapid treatment and that are widely known to critical care and emergency practitioners. The specific treatment bundles were not proposed until 2012, but the components of the 3-hour bundle represent standard elements of excellent sepsis care and have always been present in well-treated patients. ...” on page 5 

- the “definition of suspected infection was based on clinicians’ perceptions of infection” on page 15, is now integrated in one of the inclusion criteria “presence of a suspected infection is based on definition following clinicians’ perceptions of infection, which was defined as a body fluid culture ordered and anti-infective administered within four hours of one another” described on page 5.

However, the “We hypothesize that patients with abnormal GCS may slow sepsis treatment, because they first receive …” in page 14, was not something we would have known at the design stage. This is one of the findings of the study that we interpreted from a clinical point of view.

Method:

Authors used XGboost and SHAP packages in R. Since these are well-known methods, I would have appreciated if you included more details about the modeling process, certain data normalization/standardization procedures for continuous variables, tuning and parameter selection phase, and the reason for not having a testing set (authors used 70% training, and 30% validation sets, split randomly). I liked the brief juxtaposition of XGBoost and logistic regression on page 13. It is better to have it earlier in the methods section. Since authors used historical data, it will be interesting to see the distribution of different Diagnosis Related Groups (DRG) or ICD-10 codes among Sepsis population. This can further validate your Sepsis identification method.

Response: We have added more technical details regarding modeling process, data preprocessing and tuning in the Method section (page 6). Note that to retain better interpretation of modeling results as well as attribute to the robustness of tree-based models such as GBM, we didn’t perform additional normalization/standardization procedures for continuous variables, except for missing value handling. We’ve also mentioned the comparison with logistic regression ahead of time.

With regard to ICD coding, this could, indeed be interesting, but probably does not validate the technique. In fact we used a method of sepsis identification very similar to that of Rhee, et al, (JAMA, 2017). Additionally, we demonstrated in a related dataset from our institution that ICD codes are quite insensitive for identifying patients with infection-induced organ dysfunction (Deis, et al. CHEST, 2017) The work would be interesting, but we don’t believe that it would enhance our findings. 

Results/Discussion:

Seems like variables related to vital signs play a big role in rapidity of treatment, which is quite expected. Authors also discuss the possibility of using other variables such as machine generated features (e.g. mean respiratory rate over time in page 15), and social factors (e.g. income, education in page 15). Since the latter is the study limitation, I would like to see what the findings of previous research about these features were and how your findings compare to them. In doing so, you may refer to some of the recent similar works, e.g. [1], [2].

Moreover, authors mentioned an important aspect of the subject where operational factors contributing to the ED wait times (e.g. staffing ratio, wait time in page 15) can impact physicians’ decision to expedite treatments. As far as I know, congested downstream units (e.g. ICU) prolongs the expected ED boarding time. Providers, on the other hand, initiate/expedite the treatment in the ED when they foresee a long wait time. I would like to see a brief discussion on this matter. Please refer to [3] for the case of ICU beds. Also, you may define the terms “risk factor” and “contributing factor” that were used for features between two different groups.

Response: We thank reviewers for this comment. [1] suggested the importance of personalized medicine advancement in helping better diagnose and treat sepsis patients, which is exactly something we are trying to achieve using a robust machine learning model. However, [1] emphasized on discovering genomic biomarkers, while we focused more on phenotypic features that are commonly recorded in EMR data. The majority of our findings involve vital signs. Although that the information might have been anecdotally predicted, we believe this to be the first study quantifying the associations of vital signs data with rapidity of treatment on a continuous scale. In [2], it pointed out that “most studies reported on changes in compliance with bundle elements and/or mortality rates”, however, few of those studies took a step further to understand risk factors that affect the compliance or non-compliance, which is something we are trying to uncover here. To emphasize on this point, we have added the following paragraph in the discussion section: “As discussed in [2], awareness and compliance with protocols can be difficult to maintain in the absence of an effective clinical pathway, which requires careful planning and dedicated resources. With a better understanding of alerting features associated with rapid treatment would provide useful and evidence-based insights to a better design for creating such clinical pathway and continuous education. ”. 

It is an interesting proposition that treatment is initiated or expedited in the ED when ED personnel foresee a long wait time. This would be true under ideal circumstances; however, in the real time workflow of the emergency department and the hospital floors and ICUs, it is rare that one can actually foresee the circumstance. Beds can and do open at any time, and patients are accepted by the admitting team, whether on the hospital ward or in the ICU, as soon as they can be. It is rare, certainly in our hospital, that the ED physicians can actually anticipate that there will be a long delay in transfer to a floor bed or ICU bed. The variable and unpredictable nature of bed availability leads to variable behavior on the part of ED physicians. The simulation you reference [3] necessarily assumes that there is a rational, predictable behavior that simply does not exist under live circumstances. In fact, a full ICU can and does lead to later implementation of sepsis bundles, once time has passed and the ED physician realizes that the patient will not be moving soon from the ED. Delays in this circumstance can be significant. 

Minor Comments:

- Page 5. (ref. to SSC document) needs a proper citation

- Page 11. “The marginal effects of the top 20 features for each model are shown in Figure 5 and Figure 6. is repeated a few lines above.

- Page 12, explanations for Figures 5 and 6 are almost identical, so I would merge them together.

- Quality of the provided figures are poor. Maybe this is because of the word-pdf conversion.

Response: We thank reviewer for the careful review! We have addressed all the minor comments and included figures of better resolution. In terms of the pictures, we have tried to save raw format as .tiff files which should guarantee better resolutions. Please also note that we have merged Figure 5 and 6 as new Figure 5, and make Figure 7 as new Figure 6. 

References:

[1] P. Palma and J. Rello, “Precision medicine for the treatment of sepsis: recent advances and future prospects,” Expert Rev. Precis. Med. Drug Dev., vol. 4, no.4, pp. 205–213, Jul. 2019.

[2] J. W. Uffen, J. J. Oosterheert, V. A. Schweitzer, K. Thursky, H. A. H. Kaasjager, and M. B. Ekkelenkamp, “Interventions for rapid recognition and treatment of sepsis in the emergency department: a narrative review,” Clin. Microbiol. Infect.,Feb. 2020.

[3] I. Hasan, E. Bahalkeh, and Y. Yih, “Evaluating intensive care unit admission and discharge policies using a discrete event simulation model,” Simulation, p.003754972091474, Apr. 2020.

---

## [Decision Letter · Decision Letter 1]

13 Jan 2021

PONE-D-20-06811R1

Clinical Factors Associated with Rapid Treatment of Sepsis

PLOS ONE

Dear Dr. Song,

Thank you for submitting your manuscript to PLOS ONE. After careful consideration, we feel that it has merit but does not fully meet PLOS ONE’s publication criteria as it currently stands. Therefore, we invite you to submit a revised version of the manuscript that addresses the points raised during the review process.

The paper was improved meaningfully, however, please follow the additional reviewers comments, and refer with a cover letter describing the corresponding modifications.

We look forward to receiving your revised manuscript.

Kind regards,

Robert Moskovitch

Academic Editor

PLOS ONE

Reviewers' comments:

Reviewer's Responses to Questions

**Comments to the Author**

1. If the authors have adequately addressed your comments raised in a previous round of review and you feel that this manuscript is now acceptable for publication, you may indicate that here to bypass the “Comments to the Author” section, enter your conflict of interest statement in the “Confidential to Editor” section, and submit your "Accept" recommendation.

Reviewer #1: (No Response)

Reviewer #2: All comments have been addressed

Reviewer #3: All comments have been addressed

2. Is the manuscript technically sound, and do the data support the conclusions?

Reviewer #1: Yes

Reviewer #2: Yes

Reviewer #3: Yes

3. Has the statistical analysis been performed appropriately and rigorously? 

Reviewer #1: Yes

Reviewer #2: Yes

Reviewer #3: Yes

4. Have the authors made all data underlying the findings in their manuscript fully available?

Reviewer #1: No

Reviewer #2: No

Reviewer #3: No

5. Is the manuscript presented in an intelligible fashion and written in standard English?

Reviewer #1: Yes

Reviewer #2: Yes

Reviewer #3: Yes

6. Review Comments to the Author

Reviewer #1: * The figure's resolution is much better now.

* Please don't use acronyms before defining them, such as ED in the abstract. Don't assume all the readers familiar with these acronyms, even if there are popular.

* Add similar research with SHAP and Sepsis or other complications. Also, studies for predicting Sepsis or other complications. Write more about it in the introduction. Compare and discuss the results with other studies, write on the differences.

Here some suggestions for quite similar studies:

Murugesan, I., Murugesan, K., Balasubramanian, L., & Arumugam, M. (2019, September). Interpretation of Artificial Intelligence Algorithms in the Prediction of Sepsis. In 2019 Computing in Cardiology (CinC) (pp. Page-1). IEEE.

Lauritsen, S. M., Kristensen, M., Olsen, M. V., Larsen, M. S., Lauritsen, K. M., Jørgensen, M. J., ... & Thiesson, B. (2020). Explainable artificial intelligence model to predict acute critical illness from electronic health records. Nature communications, 11(1), 1-11.

Yang, M., Liu, C., Wang, X., Li, Y., Gao, H., Liu, X., & Li, J. (2020). An Explainable Artificial Intelligence Predictor for Early Detection of Sepsis. Critical Care Medicine, 48(11), e1091-e1096.

Komorowski, M., Celi, L. A., Badawi, O., Gordon, A. C., & Faisal, A. A. (2019). Understanding the Artificial Intelligence Clinician and optimal treatment strategies for Sepsis in intensive care. arXiv preprint arXiv:1903.02345.

And similar papers for prediction of complications:

Reyna, M. A., Josef, C., Seyedi, S., Jeter, R., Shashikumar, S. P., Westover, M. B., ... & Clifford, G. D. (2019, September). Early prediction of Sepsis from clinical data: the PhysioNet/Computing in Cardiology Challenge 2019. In 2019 Computing in Cardiology (CinC) (pp. Page-1). IEEE.

Fleuren, L. M., Klausch, T. L., Zwager, C. L., Schoonmade, L. J., Guo, T., Roggeveen, L. F., ... & Elbers, P. W. (2020). Machine learning for the prediction of Sepsis: a systematic review and meta-analysis of diagnostic test accuracy. Intensive care medicine, 1-18.

Itzhak, N., Nagori, A., Lior, E., Schvetz, M., Lodha, R., Sethi, T., & Moskovitch, R. (2020, August). Acute Hypertensive Episodes Prediction. In International Conference on Artificial Intelligence in Medicine (pp. 392-402). Springer, Cham.

Mao, Q., Jay, M., Hoffman, J. L., Calvert, J., Barton, C., Shimabukuro, D., ... & Das, R. (2018). Multicentre validation of a sepsis prediction algorithm using only vital sign data in the emergency department, general ward and ICU. BMJ open, 8(1).

Cherifa, M., Blet, A., Chambaz, A., Gayat, E., Resche-Rigon, M., & Pirracchio, R. (2020). Prediction of an acute hypotensive episode during an ICU hospitalization with a super learner machine-learning algorithm. Anesthesia & Analgesia, 130(5), 1157-1166.

Reviewer #2: (No Response)

Reviewer #3: In the revised manuscript titled "Clinical Factors Associated with Rapid Treatment of Sepsis", the authors improved dramatically the writing. It is now much clearer, and the authors addressed all previous comment and suggestion raised by the reviewer.

However I still have a few comments which can improve the manuscript farther.

- The author listed 3 criteria "Patients were included by satisfying the following criteria:" Are all three criteria required for every sample to be included?

- "EMS personnel" abbreviation is used w/o definition.

- Table 2:

1. There is no need for replicating column titles for every subsection.

2. I would recommend to stratify columns also by rapid treatment vs not.

- Figure 6: It is not clear whether this is for group 2 only, or cross sectional.

7. PLOS authors have the option to publish the peer review history of their article (what does this mean?). If published, this will include your full peer review and any attached files.

Reviewer #1: No

Reviewer #2: No

Reviewer #3: No

---

## [Author Response · Author response to Decision Letter 1]

5 Mar 2021

Response to Reviewer 1: We appreciate reviewer’s comments for the minor revision. 

- We have identified all acronyms and added their full spellings when first introduced in the text. In addition, we have added a new “Acronym” section at the beginning of the text. 

- We have also enriched the “Introduction” section with all the recommended articles added. On second page, we added “…Numerous efforts have been made to improve prognostic accuracy and efficiency for sepsis and its complications via machine learning techniques. For example, Yang et al.(8), Komorowski et al. (9) and Reyna et al. (10) developed artificial intelligence models to predict sepsis in intensive care; while Mao et al. (11) and Lauritsen et al. (12) extended the prediction application to ED and general ward. Itzhak et al. (13) and Cherifa et al. (14) developed models to predict acute hypertensive or hypotensive episodes among ICU admissions. However, few, if any studies have been designed to understand specific clinical features that patients exhibit at the time that physicians initiate rapid treatment of sepsis. Without assuming causality, one could evaluate from a situational awareness perspective which clinical features are most closely associated with rapid, thorough treatment. We believed that data from such a study could provide novel information that could be used to prompt rapid sepsis treatment for appropriate patients, regardless of the extant diagnostic criteria.”

Response to Reviewer 3: We appreciate reviewer’s comments for the minor revision.

- Yes. All three criteria are required for the patient encounter to be eligible for the study, and we have stated so very specifically in this revision. 

- EMS stands for Emergency Medical Service, besides EMS, we have identified all acronyms and added their full spellings when first introduced in the text. In addition, we have added a new “Acronym” section at the beginning of the text.

- For Table 2, we believe we understand what the reviewer is intending, and we considered altering the table. However, on reflection, we believe that we should leave the table as is. The main thrust of the paper is to identify the demographic and other characteristics that identify, i.e. separate, the patients who receive rapid treatment, compared with those who do not. We do so with much more sophisticated techniques than one would employ with a perusal of summary statistics in a table. We opted to provide the demographics of the entire study set and to look at the subsets of those who required vasopressors and those who did not, and we think this better serves the purpose of informing the reader of who our study population is. This is underscored by the fact that demographic information was not among the features with the greatest effect in the model. The physiological data that most significantly separate rapidly treated patients from others are displayed in Figure 2, while demographics do not rise to that level of influence. We did remove the redundant column titles, as suggested. 

- Figure 6 is on the entire or cross-sectional data set, i.e. for both group 1 and group 2.

---

## [Decision Letter · Decision Letter 2]

19 Apr 2021

Clinical Factors Associated with Rapid Treatment of Sepsis

PONE-D-20-06811R2

Dear Dr. Song,

We’re pleased to inform you that your manuscript has been judged scientifically suitable for publication and will be formally accepted for publication once it meets all outstanding technical requirements.

Kind regards,

Robert Moskovitch

Academic Editor

PLOS ONE

Additional Editor Comments (optional):

Following the reviewers response, it seems that the paper is ready for publication.

Reviewers' comments:

Reviewer's Responses to Questions

**Comments to the Author**

1. If the authors have adequately addressed your comments raised in a previous round of review and you feel that this manuscript is now acceptable for publication, you may indicate that here to bypass the “Comments to the Author” section, enter your conflict of interest statement in the “Confidential to Editor” section, and submit your "Accept" recommendation.

Reviewer #1: All comments have been addressed

Reviewer #3: All comments have been addressed

2. Is the manuscript technically sound, and do the data support the conclusions?

Reviewer #1: Yes

Reviewer #3: Yes

3. Has the statistical analysis been performed appropriately and rigorously? 

Reviewer #1: Yes

Reviewer #3: Yes

4. Have the authors made all data underlying the findings in their manuscript fully available?

Reviewer #1: Yes

Reviewer #3: No

5. Is the manuscript presented in an intelligible fashion and written in standard English?

Reviewer #1: Yes

Reviewer #3: Yes

6. Review Comments to the Author

Reviewer #1: (No Response)

Reviewer #3: I see that the authors of the manuscript "Clinical Factors Associated with Rapid Treatment of Sepsis" addressed all reviewers' comments adequately.

7. PLOS authors have the option to publish the peer review history of their article (what does this mean?). If published, this will include your full peer review and any attached files.

Reviewer #1: No

Reviewer #3: No

---

## [Editor Report · Acceptance letter]

26 Apr 2021

PONE-D-20-06811R2 

Clinical Factors Associated with Rapid Treatment of Sepsis 

Dear Dr. Song:

I'm pleased to inform you that your manuscript has been deemed suitable for publication in PLOS ONE. Congratulations! Your manuscript is now with our production department. 

Kind regards, 

on behalf of

Dr. Robert Moskovitch 

Academic Editor

PLOS ONE